# Targeting NAD+ regeneration enhances antibiotic susceptibility of *Streptococcus pneumoniae* during invasive disease

Hansol Im[1], Madison L. Pearson[1], Eriel Martinez[1], Kyle H. Cichos[2], Xiuhong Song[1], Katherine L. Kruckow[1], Rachel M. Andrews[1], Elie S. Ghanem[2], Carlos J. Orihuela[1] *

1 Department of Microbiology, Heersink School of Medicine, The University of Alabama at Birmingham, Birmingham, Alabama, United States of America, 2 Department of Orthopaedic Surgery Arthroplasty Section, Heersink School of Medicine, The University of Alabama at Birmingham, Birmingham, Alabama, United States of America

* corihuel@uab.edu

**Data Availability Statement:** All relevant data are within the paper and its Supporting Information files

## Abstract

Anaerobic bacteria are responsible for half of all pulmonary infections. One such pathogen is *Streptococcus pneumoniae* (*Spn*), a leading cause of community-acquired pneumonia, bacteremia/sepsis, and meningitis. Using a panel of isogenic mutants deficient in lactate, acetyl-CoA, and ethanol fermentation, as well as pharmacological inhibition, we observed that NAD(H) redox balance during fermentation was vital for *Spn* energy generation, capsule production, and in vivo fitness. Redox balance disruption in fermentation pathway-specific fashion substantially enhanced susceptibility to killing in antimicrobial class-specific manner. Blocking of alcohol dehydrogenase activity with 4-methylpyrazole (fomepizole), an FDA-approved drug used as an antidote for toxic alcohol ingestion, enhanced susceptibility of multidrug-resistant *Spn* to erythromycin and reduced bacterial burden in the lungs of mice with pneumonia and prevented the development of invasive disease. Our results indicate fermentation enzymes are de novo targets for antibiotic development and a novel strategy to combat multidrug-resistant pathogens.

## Introduction

Anaerobic bacteria are responsible for more than half of pleuropulmonary infections including lung abscesses and pneumonia [1]. Anaerobic bacteria are part of the normal flora of mucosal membranes and the diseases they cause often reflect outgrowth from the site where they are colonized [2]. One such pathogen is *Streptococcus pneumoniae* (*Spn*), the leading cause of community-acquired pneumonia (CAP) [3]. In most instances, *Spn* resides asymptomatically in the human nasopharynx; however, this gram-positive opportunistic pathogen can cause sinusitis and otitis media, as well as life-threatening infections such as pneumonia, bacteremia, and meningitis [4,5]. Globally, more than 3 million individuals are hospitalized due to pneumococcal disease annually, and hundreds of thousands die as a result [6]. Conjugate vaccines, composed of the bacterium's polysaccharide capsule conjugated to a carrier protein, are widely

**Funding:** This work was supported by National Institute of Health (AI114800, AI148368, AI156898, and AI172796 to C.J.O; and HL129948 to H.I.). The funders had no role in study design, data collection and analysis, decision to publish, or preparation of the manuscript.

**Competing interests:** The authors have declared that no competing interests exist.

**Abbreviations:** Adh/AdhE, alcohol dehydrogenase; ANOVA, analysis of variance; CAP, community-acquired pneumonia; CSP-2, competence-stimulating peptide variant 2; FM, fomepizole; GAS, Group A streptococci; GBS, Group B streptococci; HRP, horseradish peroxidase; LAB, lactic acid bacteria; Ldh, lactate dehydrogenase; MIC, minimal inhibitory concentration; PdhC, pyruvate dehydrogenase complex; SHX, serine hydroxamate; *Spn, Streptococcus pneumoniae*; THY, Todd–Hewitt broth with 0.5% yeast extract; WT, wild type.

distributed and effective in preventing disease against the included serotypes [7]. Unfortunately, the most recently licensed versions of these vaccines carry only 15 and 20 of the >100 capsule types carried by *Spn*, and a considerable gap in coverage exists [8,9]. Contributing to the urgency of the situation, epidemiological studies indicate that multidrug-resistant nonvaccine serotypes of *Spn* are emergent [10–13]. Indeed, a recent systematic analysis on the global burden of bacterial antimicrobial resistance reported that *Spn* is the fourth leading cause of death associated with antibiotic resistance. Notably, the authors stated that *Spn*'s most prevalent antibiotic resistance was against penicillin, macrolides, and Trimethoprim/Sulfamethoxazole, all of which are used commonly to treat bacterial infection [14]. Thus, the discovery of new strategies to combat drug-resistant *Spn* infection is vital.

*Spn* is a member of the lactic acid bacteria (LAB) group [15]. LAB rely exclusively on glycolysis and fermentation for energy production [16]. During fermentation, pyruvate is converted to lactate, acetate, and ethanol [17], and NADH is oxidized to regenerate NAD+, which is needed for glycolysis [17]. Accordingly, maintenance of an available NAD+ pool, i.e., redox balance, is vital for sustained energy production, bacterial growth, and survival [18,19]. In *Spn*, fermentation is driven by the enzymes lactate dehydrogenase (Ldh), alcohol dehydrogenases (Adh/AdhEs), and the pyruvate dehydrogenase complex (PdhC). These convert pyruvate to lactate, ethanol, and acetyl-CoA, respectively. Consistent with their importance, investigators have shown that *Spn* mutants deficient in *adhE* and *ldh* are attenuated for growth and virulence versus their parental controls [20,21].

Herein, we evaluated whether disruption of bacterial redox balance could serve as means to block disease caused by *Spn*. To do so, we examined the impact of mutations in genes involved in fermentation and NAD+ regeneration, as well as treatment of *Spn* with a competitive inhibitor of alcohol dehydrogenases, 4-methylpyrazole (fomepizole (FM)) [22,23], on *Spn* physiology, fitness, and susceptibility to concurrent antibiotic pressure in vitro and in vivo. Our results indicate that the blocking of NAD+ regeneration pathways during infection is a way to increase antibiotic susceptibility in drug-resistant gram-positive anaerobic pathogens, and this has clinical potential with regard to microbial eradication and treatment of disseminated infection.

## Results

### NAD+/NADH imbalance alters the *Spn* energy pool

We explored the importance of redox balance in *Spn* by testing the impact of disrupted fermentation on energy metabolism and growth using genetic mutants. We created isogenic mutants of *Spn* deficient in both of its alcohol dehydrogenases (Δ*adh*, Δ*adhE*), the pyruvate dehydrogenase complex (Δ*pdhC*), and lactate dehydrogenase (Δ*ldh*). We also tested a mutant lacking NADH oxidase (Δ*nox*), as Nox is involved in NAD+ regeneration independent of fermentation (Fig 1A). To ensure that the phenotypes observed for our panel of mutants were specific for the affected loci, we also created a panel of revertant mutants with the target loci restored.

Mutation of *adh*, *adhE*, and *nox* resulted in a 58.6%, 29.3%, and 22.1% reduction in total NAD(H) levels, respectively (Fig 1B). Moreover, the ratio of NAD+/NADH, an indicator of impaired glycolysis, was increased by 44.4%, 11.1%, and 20.4%, respectively. The Δ*ldh* mutant showed no changes in total NAD(H) pool, but a 57.4% increase in the NAD+/NADH ratio. There was also no change in the size of the NAD(H) pool for Δ*pdhC*; however, we observed a 20% decrease in the NAD+/NADH ratio, a finding that suggests the Δ*pdhC* mutant was more efficient at this process. Subsequent measurement of intracellular ATP revealed that these mutations impacted intracellular ATP concentration (Fig 1C). Stark decreases were observed

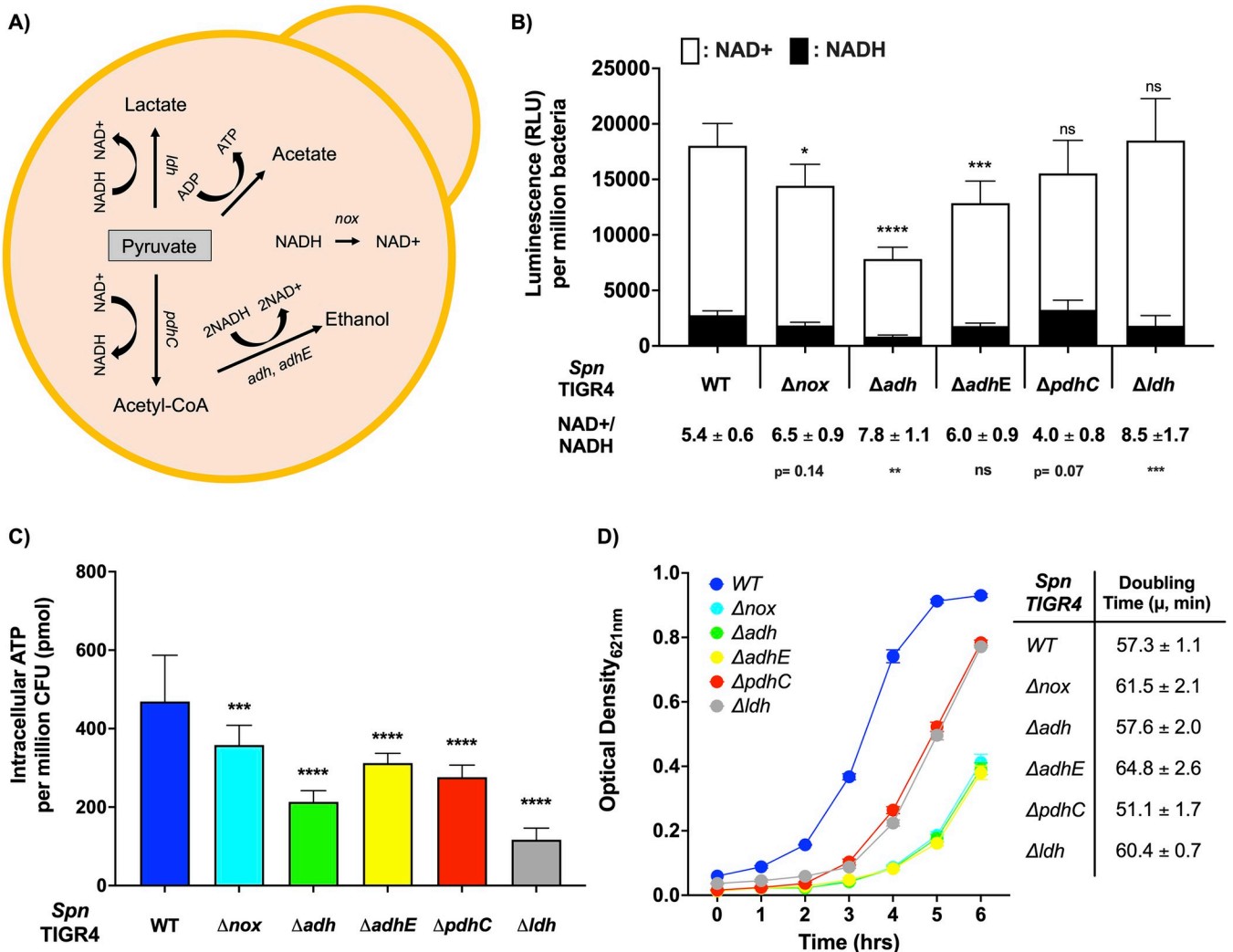

**Fig 1. Influence of NAD+ regeneration on *Spn* physiology.** (**A**) Scheme of NAD+/NADH-associated metabolism in *Spn* and the targets tested in this study by mutation: *ldh*, lactate dehydrogenase; *adh* and *adhE*, alcohol dehydrogenase; *pdhC*, pyruvate dehydrogenase complex; *nox*, NADH oxidase. (**B**) Levels of NAD (H) measured in designated *Spn* and the ratio of NAD+/NADH ($n = 3$). (**C**) Intracellular ATP concentration and (**D**) growth curve of mutant *Spn* and wild-type (WT) control ($n = 4$). ATP concentration was measured during exponential phase of growth ($OD_{620} = 0.4$). The data underlying Fig 1B, 1C and 1D can be found in S1 Data. Statistical analyses were performed using a one-way ANOVA with Dunnett post hoc test. Asterisks indicate statistical significance: *, $P \leq 0.05$, **, $P \leq 0.01$; ***, $P \leq 0.001$, ****, $P \leq 0.0001$.

for Δ*ldh* and Δ*adh* mutants, which resulted in a 54.5% and 75.1% reduction in the ATP pool versus wild type (WT), respectively. Also significant, but more modest, were decreases in the ATP pool for Δ*nox*, Δ*adhE*, and Δ*pdhC*. All 5 mutants exhibited extended lag phase during growth in media (Fig 1D). Mutants deficient in *adh*, *adhE*, and *nox* were most negatively impacted, whereas those deficient in *ldh* and *pdhC* had an intermediate phenotype. Doubling time was not impaired once bacteria reached exponential phase, with Δ*pdhC* observed to replicate faster than WT. Importantly, all of the revertant strains had phenotypes equivalent to WT, indicating that observations made with our mutants were not due to off-target effects (S1 Fig). Finally, to further dissect the divergence between the observed extended lag phase and unimpaired exponential phase of our mutants, we measured metabolic activity using heat flow [24]. With exception to the *ldh* mutant, none of the mutants in our panel displayed changes in

maximal heat level (S2 Fig). This suggests that under the ideal growth conditions provided by media, the nonaffected fermentation pathways are compensatory.

## NAD+/NADH redox imbalance influences the production of *Spn* virulence factors and in vivo fitness

Carbon availability has been shown to influence the 3 fermentation pathways and the production of pneumococcal virulence determinants [25], especially capsular polysaccharide [24]. We observed that all *Spn* mutants in our panel produced significantly less capsular polysaccharide than WT. Levels of capsule were determined by immunoblot using antibody against serotype 4 capsule and by measuring the exclusion zone around *Spn* in suspension with FITC-dextran (Fig 2A) [24]. When we measured pneumolysin, *Spn*'s pore-forming toxin, by western blot, *Δldh* was also observed to have a negative impact on its production (Fig 2B). Whereas multiple investigators have examined the individual effect of disrupted fermentation on *Spn* virulence, head-to-head comparisons of the impact of these mutations is lacking. Following intranasal challenge with a 1:1 ratio of WT and mutant pneumococci, we enumerated bacterial titers in the nasopharynx of asymptomatically colonized mice. Consistent with the notion that *Spn* with reduced levels of capsule are more fit for colonization [26], mutants outcompeted WT over the course of the 2-week colonization period (S3 Fig). Also, as expected, we observed the opposite result following forced aspiration of mice in a pneumonia model. In this instance, and at an anatomical site where the lack of capsule would be a liability, there was a very strong attenuating effect for the mutants, with only WT *Spn* recoverable from the majority of mice. One key exception to this, and consistent with its enhanced growth rate in vitro, the *ΔpdhC* mutant outcompeted the WT strain in the lungs of mice (Fig 2C). Our conclusion from these results is that alteration of *Spn* metabolism and redox balance has profound effects on virulence, and this can be decreased or increased depending on the anatomical site tested and pathway targeted.

## NAD+/NADH redox imbalance influences susceptibility to antibiotic stresses

Antibiotics, depending on their mechanism of action, impose distinct forms of stress. This can lead to dormancy, i.e., bacteriostatic antimicrobials, and death, i.e., bactericidal antimicrobials [27,28]. It was unclear how disruption of *Spn*'s central metabolism would influence the efficacy of different antimicrobial agents. We therefore examined the susceptibility of our mutant panel to 3 different groups of antibiotics; those targeting ribosomal activity: erythromycin, gentamicin, and chloramphenicol [29]; those inhibiting DNA transcription: rifampicin [30]; and those disruptive of cell wall synthesis: penicillin and vancomycin [31]. For the study, viable log-phase bacteria in media were enumerated following antimicrobial treatment (T = 0) at set intervals for up to 3 hours.

Following erythromycin treatment, we observed enhanced killing for all the mutants except *ΔadhE*. In comparison to WT, *Δldh*, *Δnox*, and *Δadh* had 8-fold to 50-fold increased susceptibility (Fig 3). For *ΔpdhC*, we observed a smaller, approximately 50%, increase in susceptibility. Following gentamicin treatment, *Δldh*, *Δadh*, *ΔadhE*, and *ΔpdhC* mutants had 7-fold to 70-fold increased susceptibility versus WT. We found that *Δnox* had no significant change in susceptibility to gentamicin. We expected to observe similar results with chloramphenicol, but only *Δldh* and *ΔpdhC* showed a slightly higher susceptibility. For penicillin, vancomycin, and rifampicin, the effect was also mutant specific, with only *Δldh* and *ΔpdhC* having enhanced sensitivity. Notably, when these mutants were exposed to penicillin and vancomycin, there was a >100-fold reduction in viability. When *Δldh* and *ΔpdhC* were exposed to rifampicin,

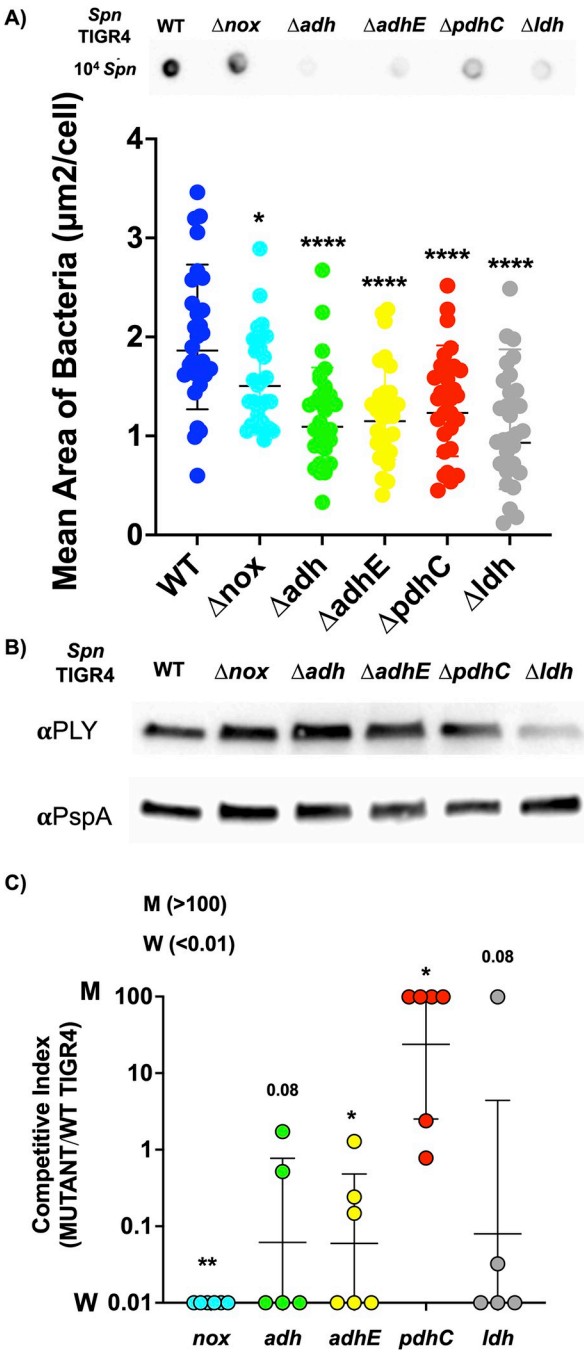

**Fig 2. Disruption of NAD+ regeneration alters *Spn* virulence factor production and in vivo fitness.** (**A**) Representative immuno dot blot for capsular polysaccharide using anti-serotype 4 antibody (done in triplicate) and relative capsule amount on the bacterial surface as determined using an FITC dextran exclusion assay. Pneumococci with greater capsule have a larger mean area per diplococcus. For each strain, pneumococci in 30 objective images were analyzed. (**B**) Representative immunoblot for pneumolysin and PspA using lab-generated polyclonal antibody. (**C**) Results of a competitive index assay for WT *Spn* (W) and designated mutants (M) following 1:1 coinfection of mice in a pneumonia model ($n$ = 5–6). Burden in the lungs at 48-hour infection was used for the comparative analysis. The data underlying Fig 2A and 2C can be found in S1 Data. Statistical analyses were done using a one-way ANOVA with Dunnett post hoc test compared to its WT. Asterisks indicate statistical significance: *, $P \leq 0.05$, **, $P \leq 0.01$; ****, $P \leq 0.0001$.

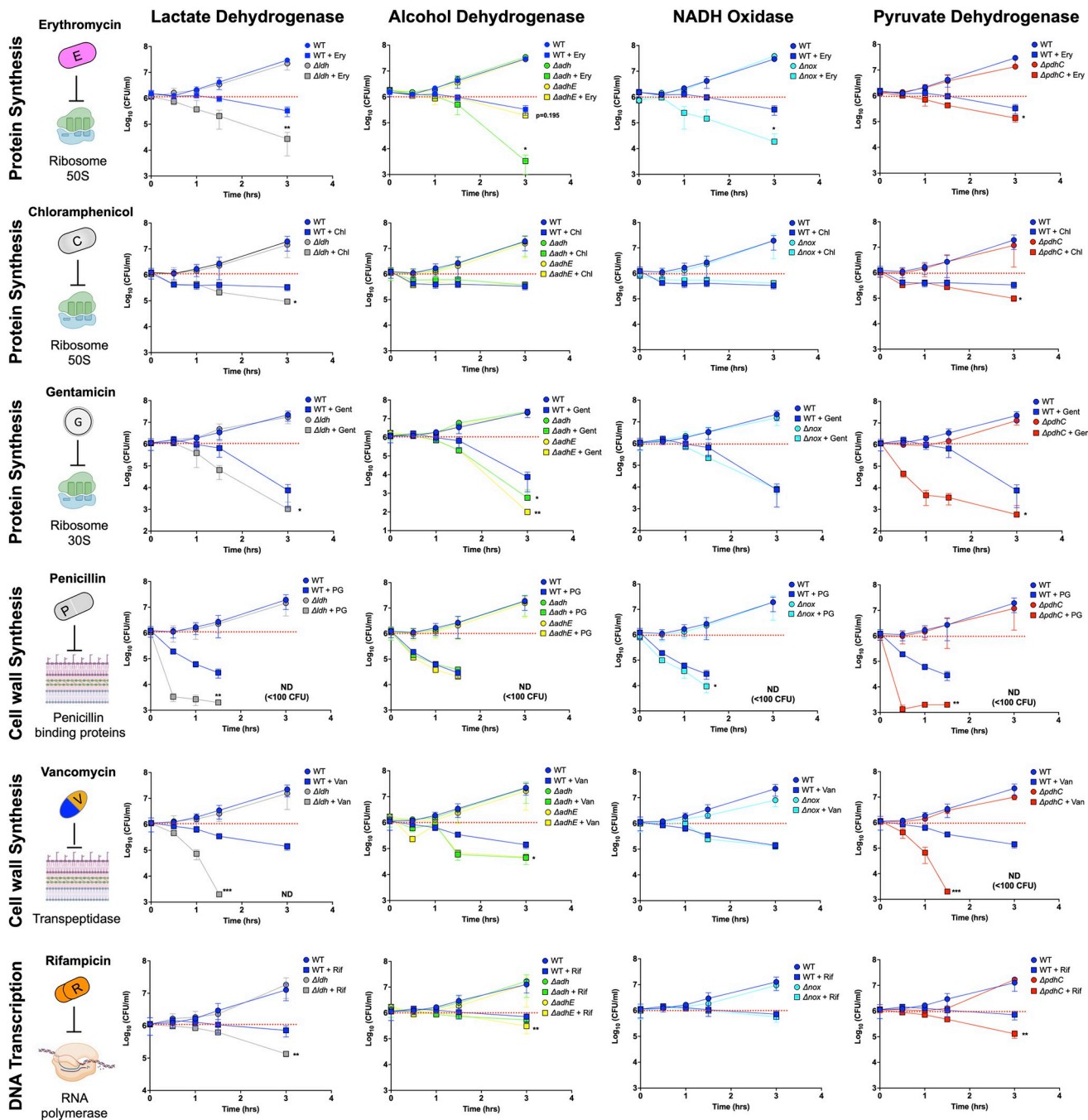

**Fig 3. Disruption of NAD+ regeneration genes influences antibiotic susceptibility.** *Spn* wild type and its isogenic mutants were enumerated following treatment with erythromycin (50 μg/ml), gentamicin (5 μg/ml), chloramphenicol (10 μg/ml), penicillin (10 μg/ml), vancomycin (10 μg/ml), and rifampicin (50 μg/ml). The data underlying this figure can be found in S1 Data. Statistics analyses was done using a Mann–Whitney *t* test with comparison made between wild type antibiotic-treated versus mutant antibiotic-treated samples ($n \geq 3$) Asterisks indicate statistical significance: *, $P \leq 0.05$, **, $P \leq 0.01$; ***, $P \leq 0.001$.

there was an 80% reduction in viability. In general, disruption of fermentation pathways enhanced susceptibility to antibiotics. While susceptibility to erythromycin was independent regardless of the fermentation pathway blocked, the rest of the tested antibiotics showed

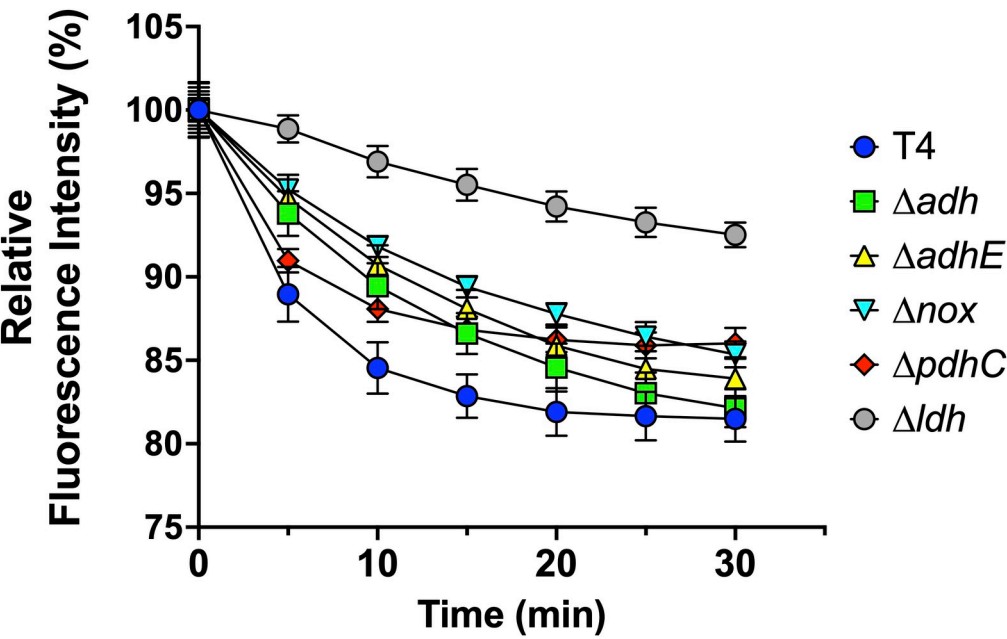

**Fig 4. Efflux capability of *Spn* TIGR4 and mutants.** Efflux pump capability of *Spn* TIGR4 WT and its isogenic mutants was performed using EtBr fluorescence measurement. The relative fluorescence level of EtBr compared to its initial time point (T$_0$), i.e., the ability to pump out the fluorescent stain, was displayed from each sample, which were monitored every 5 minutes for 30 minutes. The data underlying this figure can be found in S1 Data. (*n* = 6)

pathway-specific phenotypes. Critically, these phenotypes were absent when we tested our panel of revertants (S4 Fig). One mechanism employed by bacteria to enhance antimicrobial resistance is efflux of the antimicrobial using ATP-pumps [32]. Since this process requires energy, we speculated that metabolic disruption would therefore impair the efflux of noxious agents and was therefore a potential explanation for the enhanced susceptibility to antimicrobials. Consistent with this notion, we observed that our mutant panel of *Spn* had decreased efflux pump capability versus WT (Fig 4).

### Fomepizole treatment caused redox imbalance

Fomepizole (4-methylpyrazole) is an FDA-approved inhibitor of alcohol dehydrogenase in mammals [22,23,33] used to treat individuals with methanol poisoning (Fig 5A) [23]. Importantly, it also has reported activity in gram-negative bacteria [22,23]. We tested whether fomepizole at 50 mg/L, which corresponds to the recommended treatment concentration range [34], altered WT *Spn*'s NAD(H) pool, levels of ATP, and susceptibility to antibiotics. Fomepizole-treated *Spn* had a 20% reduction in intracellular NAD(H) levels, an approximately 95% reduction in ATP concentration, and diminished growth rate (Fig 5B and 5C). No changes in the ratio of NAD+/NADH were observed (Fig 5B). Dose dependency experiments showed meaningful depletion of ATP occurred even at the lowest concentrations of fomepizole tested, i.e., 6.25 mg/L (Fig 5C), although there was no significant impact on growth rate at concentrations below than 25 mg/kg (Table 1). Fomepizole treatment reduced capsule production by *Spn* and when we analyzed the transcription of genes involved in NAD+ regeneration, we found that fomepizole induced higher transcription of *adhE* and slightly increased but not significantly *adh* expression (S5 Fig). We interpret the latter observation as indicating that *Spn* were attempting to overcome fomepizole-induced stress.

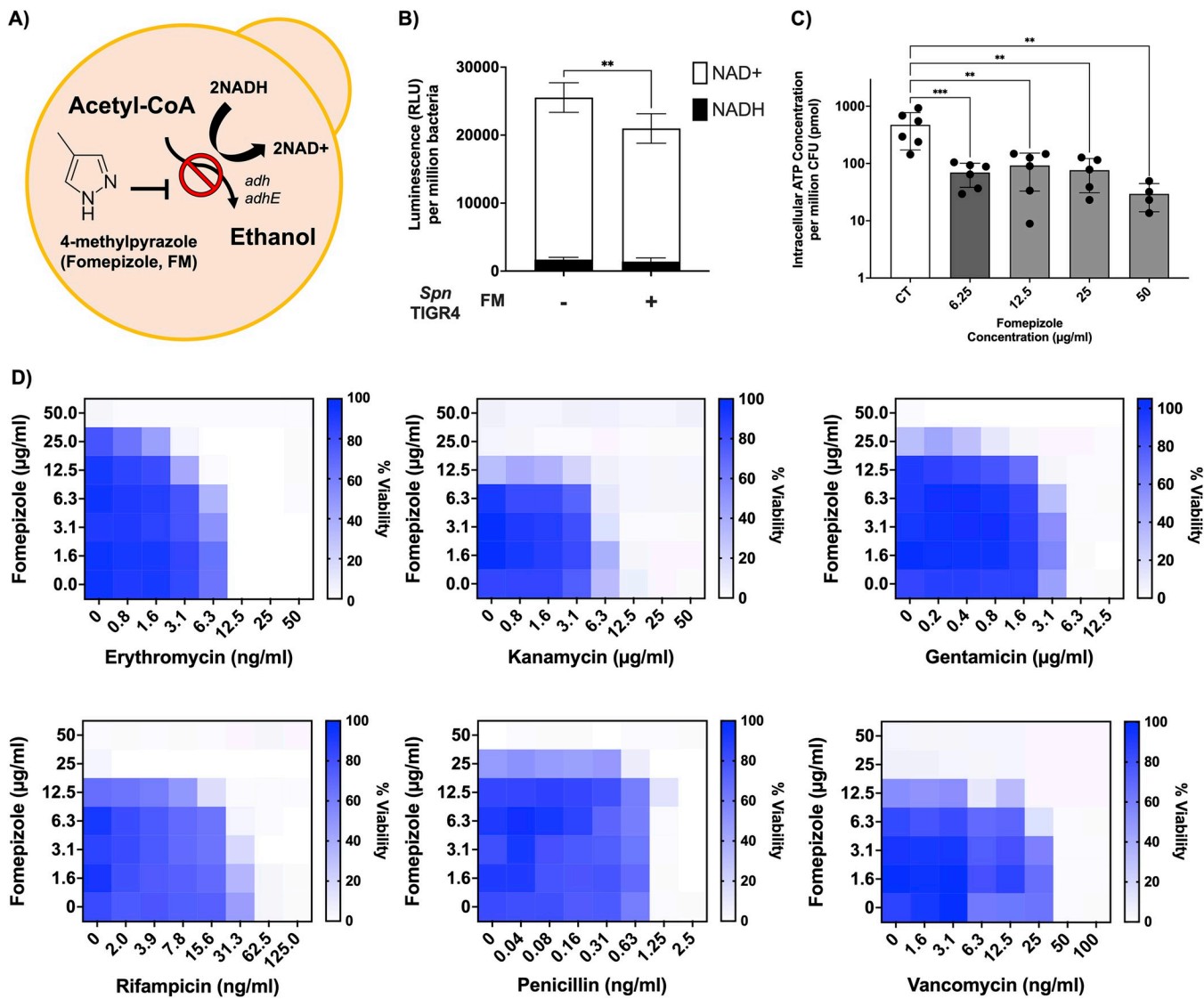

**Fig 5. Fomepizole treatment decreases the NAD(H) pool, impairs ATP production, and enhances ribosomal targeting antibiotic efficacy.** (**A**) A schematic of the fomepizole mechanism. Fomepizole treatment of *Spn* for 2 hours resulted in (**B**) alteration of the NAD(H) pool ($n \geq 4$) and (**C**) reduced intracellular ATP concentrations ($n \geq 4$). (**D**) Heat plots of microdilution checkerboard assay, an assay for bacterial viability, following addition of fomepizole and erythromycin, kanamycin, gentamicin, rifampicin, penicillin, and vancomycin. At least 3 independent assays were performed. ($n \geq 3$). The data underlying Fig 5B, 5C and 5D can be found in S1 Data. Statistical analyses were done with the Mann–Whitney *t* test and one-way ANOVA. Asterisks indicate statistical significance: *, $P \leq 0.05$; **, $P \leq 0.01$; ***, $P \leq 0.001$.

### Fomepizole treatment enhanced susceptibility of *Spn* to antibiotics

We subsequently explored the effect of fomepizole in a combinatorial treatment with antibiotics. Fomepizole, used at 12.5 mg/L to 25 mg/L, decreased the minimal inhibitory concentration (MIC) of erythromycin and gentamicin by 4-fold versus control (Table 1 and Fig 5D). We also observed a 2-fold reduction in the MIC for kanamycin and rifampicin (Table 1 and Fig 5D). No significant impact of fomepizole used below 50 mg/L was observed on susceptibility to penicillin, but a 2-fold decrease in the MIC of vancomycin was detected (Table 1 and Fig 5D). Direct antibiotic-killing assays confirmed these results. Briefly, recoverable bacteria were enumerated at regular intervals after antibiotic treatment with and without fomepizole in media.

**Table 1. MIC alteration of *Spn* TIGR4 by the combination of antibiotics and fomepizole.**

| Bacterial Strain | Compounds/Antibiotics | MIC[a] (µg/ml) | MIC[b] (µg/ml, ADJ) | |
|---|---|---|---|---|
| *Spn* TIGR4 | Fomepizole | <50 | | |
| *Spn* D39 | Fomepizole | 100 | | |
| *Streptococcus pyogenes* (GAS) | Fomepizole | >400 | | |
| *Streptococcus agalactiae* (GBS) | Fomepizole | >400 | | |
| *Enterococcus faecium* | Fomepizole | >400 | | |
| *Spn* TIGR4 | Erythromycin | 0.0125 | 0.003125 (25) | FM as ADJ (mg/kg) |
| | | 0.0125 | 0.00625 (12.5) | |
| | Kanamycin | 25 | 6.25 (12.5) | |
| | Gentamicin | 6.25 | 1.56 (25) | |
| | Penicillin G | 0.00125 | 0.00125 (25) | |
| | Vancomycin | 0.050 | 0.025 (12.5) | |
| | Rifampicin | 0.0625 | 0.03125 (25) | |
| *Spn* D39 | Erythromycin | 0.0125 | 0.00625 (50) | |
| *Streptococcus pyogenes* (GAS) | Erythromycin | 0.03125 | 0.0039 (400) | |
| | | 0.03125 | 0.0156 (>50) | |
| *Streptococcus agalactiae* (GBS) | Erythromycin | 0.025 | 0.00625 (400) | |
| *Enterococcus faecium* | Gentamicin | 50 | 12.5 (>200) | |

[a]MIC of compound/antibiotic alone.

[b]MIC of antibiotics with fomepizole.

(ADJ): Concentration of adjuvant tested.

Fomepizole treatment improved antimicrobial killing of antibiotics that inhibited protein synthesis, but no significant difference in recoverable bacteria was observed when fomepizole-treated *Spn* were exposed to cell wall targeting antibiotics (S6 Fig). We also evaluated whether fomepizole treatment impacted the antibiotic susceptibility of other anaerobic gram-positive bacteria, including other streptococcal pathogens, i.e., *Streptococcus pyogenes* (Group A streptococci (GAS)), *Streptococcus agalactiae* (Group B streptococci (GBS)), and *Enterococcus faecium*, to erythromycin or gentamicin. We observed from 2-fold to 8-fold decreased MIC with fomepizole in most cases, including *E. faecium* (S7 Fig and Table 1).

## Fomepizole works alongside erythromycin to enhance killing of *Spn* in vivo

Finally, we tested whether this increased antibiotic susceptibility to erythromycin following treatment with fomepizole held true in mice and if this combination had potential therapeutic value. For this test, we chose a multidrug-resistant clinical isolate of *Spn* serotype 35B strain 162–5678, which has high resistance to erythromycin (MIC of 12.5 µg/ml), 1,000-fold higher resistance than the laboratory strain TIGR4 used for experiments above. We also determined that *Spn* 35B 162–5678 carried *ermB*, *mef(A/E)*, and *aphA3* genes, all of which were absent in the *Spn* TIGR4 strain (S8 Fig). It is also notable that *Spn* 35B serotype has been reported as an emerging multidrug-resistant serotype in clinical settings [35]. For the in vivo test, intratracheally challenged mice received a single intraperitoneal injection of erythromycin (25 mg/kg), with or without fomepizole (50 mg/kg), at 18 hours post-infection. The concentration of erythromycin used was in the range used to treat patients, including children [36]. Infected mice

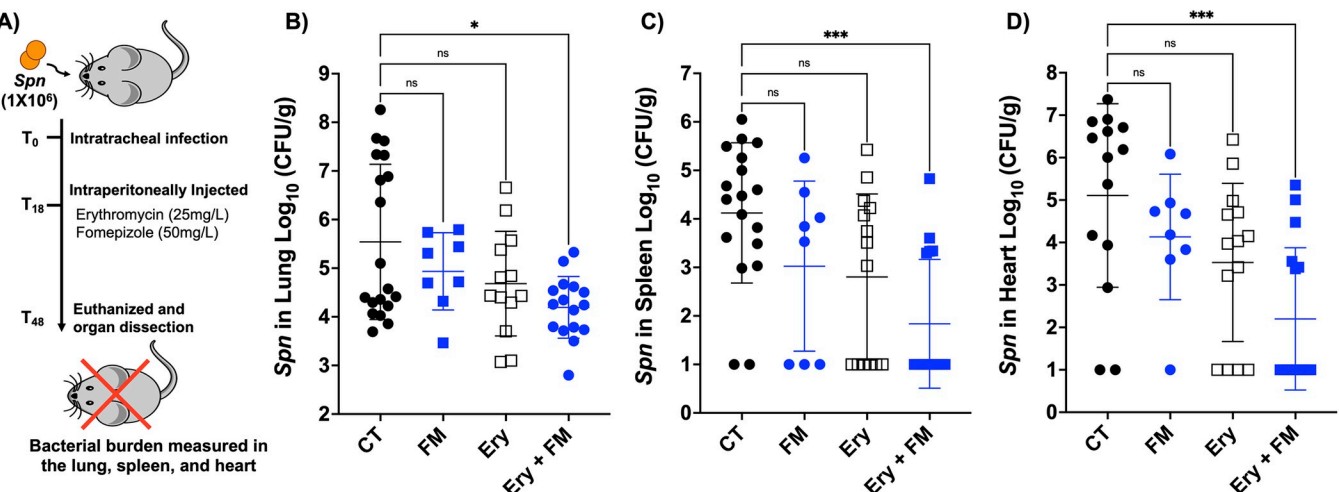

**Fig 6. Fomepizole enhances erythromycin susceptibility in multidrug-resistant *Spn* pneumonia and invasive disease.** The potential of fomepizole as an adjuvant was tested in a pneumonia model using multidrug-resistant *Spn* serotype 35B. (**A**) Infection was done intratracheally, with erythromycin and fomepizole injected intraperitoneally 18 hours post-infection. At 48 hours post-infection, animals were killed, and the bacterial burden in the (**B**) lungs, (**C**) spleen, and (**D**) heart were determined ($n \geq 8$). Statistical analyses were done using a Kruskal–Wallis ANOVA test with Dunnett post hoc test with control group. Asterisks indicate statistical significance: $^{*}$, $P \leq 0.05$; $^{**}$, $P \leq 0.01$; $^{***}$, $P \leq 0.001$.

were killed at 48 hours post-infection, and we measured bacterial burden in the lungs, spleen, and heart as indicators of disease severity (Fig 6A). We purposefully tested the impact of a single treatment dose at the early/mid stage of pneumonia so as to not eradicate the bacteria and, therefore, have sensitivity to detect intermediate effects.

Therapeutic treatment of *Spn*-infected mice with erythromycin or fomepizole alone did not change the bacterial burden measured in the lungs, spleen, or hearts of infected mice (Fig 6B–6D). However, mice receiving the combination therapy had significantly lower pneumococcal burden in all organs compared to the control group (Fig 6). In the lung, combination treatment decreased bacterial burden by approximately 95%. The difference between combination treatment and single treatment or control groups was more starkly revealed in sites of invasive disease, the spleen and heart. In the spleen, combination treatment decreased bacterial burden approximately 100-fold. When we measured bacterial burden in the heart, we found approximately 700-fold less pneumococci in the combination treatment group. Notably, only 37% (6 of the 16) of the mice receiving combination treatment had detectable bacteria in the spleen and heart. In contrast, the majority of mice in the erythromycin-treated group, 71% (10 of 14 mice) developed invasive disease. These results strongly support the potential of fomepizole and possibly other agents that target fermentation pathway and redox balance as an adjuvant therapy to antimicrobials and means to prevent the development of life-threatening invasive disease.

## Discussion

Herein, we demonstrate that disruption of *Spn* redox balance alters *Spn* energy levels, physiology, and virulence determinant production. What is more, impaired redox balance culminated in altered fitness and susceptibility to antimicrobials both in vitro and in vivo. The impact of impaired metabolic activity varied considerably depending not only on the fermentation pathway that was targeted but also on the anatomical site tested; a dramatic difference was observed between colonization and pneumonia. For the latter, the most notable phenotype was seen following the disruption of alcohol dehydrogenase activity, with the potential to dramatically

increase the efficacy of erythromycin in vivo. Our results suggest that metabolic activity is a viable target for antimicrobial targeting during infection but that this also depends on the physiological demands placed on the bacterium due to its location in the body.

Our genetic approach revealed that deletion mutants of genes involved in NAD(H) redox balance had moderate to major effects on the NAD(H) pool, intracellular ATP concentrations, and growth rates of the bacteria. These observations were consistent with the large body of published work that shows NAD+/NADH redox balance regulates metabolic status and energy production [19,37,38]. Deletion of *adh* had the most consistent negative impact on the pneumococcus, with NAD and ATP levels being severely restricted as result. Disruption of other metabolic pathways was less impactful but in general had negative consequence. A recent study by our group has shown that *adh* (SP_0285, zinc-containing alcohol dehydrogenase) is highly expressed in the airway and organs of mice during invasive pneumococcal disease [24,39]. Other investigators have reported that *adh* is a member of the CiaRH stress regulon and that its transcription is up-regulated during penicillin-induced stress [40,41]. Herein, we also showed that *adh* is required in vivo and, if blocked pharmacologically with fomepizole, starkly enhances the susceptibility to erythromycin. We also observed that deletion of *adhE* (SP_2026) significantly influenced the NAD pool and amount of available ATP.

Previously, Luong and colleagues reported that *adhE* deletion decreased the production of *Spn* choline-binding proteins and pneumolysin and was attenuating in a pneumonia model of infection [21]. Thus, our work agrees with and builds on prior work that alcohol dehydrogenases are vital enzymes. Adh (SP_0285) and AdhE (SP_2026) share near identical binding sites for their substrates [42,43]. It is noteworthy that in follow-up experiments, a double knockout of *adh/adhE* behaved similarly to the *adhE* mutant (S9 Fig). Our interpretation of this result is that any compensation for a mutation in this pathway comes primarily through other fermentation pathways. Notably, both *adh* and *adhE* mutations and fomepizole treatment inhibited capsule production, and this has not previously been described (S5 Fig).

We found that ATP levels were positively correlated with capsular polysaccharide production in our mutant panels. This observation is not surprising given the high metabolic cost to produce capsule, a highly complex glycoconjugate, and prior work by our lab and others showing that glucose, and ongoing carbon-catabolite repression, positively influences capsule production [24,44]. Capsule helps *Spn* avoid entrapment in negatively charged mucous during colonization and is required for virulence in the lower airway and bloodstream [45]. Yet, capsule levels are not static, and variants with less or more capsule are selected for depending on the anatomical site examined. More specifically, *Spn* within the nasopharynx are favored when they have less capsule, which reveals their adhesins [26]. *Spn* within the bloodstream are favored when they have more capsule, which blocks interactions with phagocytes [24]. We propose it is for this reason that our mutant panel was favored in the upper airway but drastically attenuated in the lungs. Pneumolysin also plays an important role during pneumonia [46], and its reduced production following *ldh* deletion is another explanation for why this mutant was attenuated following intratracheal challenge.

Results with the *pdhC* mutant were consistently an exception. In 2020, Echlin and colleagues found that deletion of *pdhC* severely reduced capsule production [47]; our results confirmed this. Echlin also showed that mutation of *pdhC* severely inhibited virulence during pneumonia, which was opposite to our own observations that instead showed the *pdhC* mutant outcompeted its parental strain in vitro and in vivo. One explanation for this difference is the distinct genetic backgrounds of the bacteria used in both studies; ours used a serotype 4 TIGR4 genetic background, whereas Echlin used a serotype 2 D39 genetic background [48–50]. Another possible explanation is the biochemical composition of the capsule and the metabolic demands imposed by its production. Accordingly, some isolates gain virulence when the

capsule is switched and other lose virulence [51]. It is possible that *pdhC* mutation imposes greater stress on bacteria that are producing serotype 2 capsule versus serotype 4. Likewise, the genome of the bacterium may not accommodate the stress imposed by *pdhC* deletion equally. Nonetheless, our work adds to the existing body of knowledge that suggests disruption of central metabolism, and redox balance has profound effects on bacterial fitness, virulence determinant production, and the ability to cause disease. Our study goes beyond current knowledge by exploring whether targeted disruption of these pathways might be leveraged to enhance antimicrobial susceptibility and, in turn, used to treat infections caused by resistant strains.

It is well documented that the metabolic state of bacteria strongly influences antibiotic efficacy in vivo [52]. Yang and colleagues used malate to enhance the activity of the TCA cycle in *Vibrio alginolyticus* and increase colistin efficacy within zebrafish [53,54]. In 2019, Li and colleagues showed that treatment of *E. coli* with L-tryptophan enhanced the metabolic rate, prevented the formation of persister cells, and increased antibiotic susceptibility [55]. Conversely, nutrient restriction or reduced metabolism can increase bacterial antibiotic resistance. For instance, Anderi and colleagues showed that *Klebsiella pneumoniae* under nutrient-restricted conditions was intrinsically resistant to killing by ampicillin and ciprofloxacin [56]. Nguyen and colleagues also demonstrated that stringent response induced by serine hydroxamate (SHX) in *Pseudomonas aeruginosa* increased antibiotic resistances [57]. Recently, Lobritz and colleagues showed that forced slowdown of *E. coli* metabolism with erythromycin impaired the activity of norfloxacin, a DNA replication–inhibiting quinolone [52]. However, Lapinska and colleagues reported opposite findings and showed that fast-growing bacterial phenotypes were more resistant to macrolide by increasing ribosomal promoter activity [58]. In general, we observed that genetic disruption of central metabolism enhanced susceptibility to antimicrobial killing. We propose the reason for this is that *Spn*'s metabolic resiliency was compromised by our disruption of its fermentation pathways. Disruption of any one pathway was on its own not sufficient to disrupt growth under favorable conditions, i.e., in bacterial media. However, and when under antibiotic pressure, the loss of energy due to our metabolic disruption impaired the bacterium's ability to overcome this stress. Since antimicrobials impose different forms of physiological stress on the bacteria dependent on their class, this explains why the observed susceptibility to killing was not uniform and varied based on the metabolic pathway that was disrupted. We further propose that in vivo, the changes in virulence determinant production that occur as a result of this loss of energy, for instance, reduced capsule, simultaneously enhance the bacterium's susceptibility for eradication by the host.

Fomepizole inhibits alcohol dehydrogenase function by acting as a competitive inhibitor. It has an 8,000-fold higher affinity to the enzyme than its substrates, i.e., ethanol [23,33] and is reported to have negligible side effects [59]. Its affinity for the bacterial versions of the enzyme is not known, but it has demonstrated efficacy in their blocking [22,60]. In fomepizole-treated *Spn*, the NAD(H) pool and intracellular ATP concentration were accordingly reduced, and erythromycin susceptibility was increased by more than 5-fold in vivo. As discussed above, we propose that the impact of fomepizole on *Spn* in vivo is most likely due to its pleiotropic effects on the bacteria: i.e., reducing available energy, induction of redox imbalance, increased antibiotic susceptibility, and loss of polysaccharide capsule, which together sensitized the bacterium for both antimicrobial and host killing [61,62]. Accordingly, we propose that blocking alcohol dehydrogenase activity enhanced susceptibility to erythromycin killing in vivo as it reduced the resiliency of the bacteria to a number of stressors while simultaneously decreasing its virulence.

Importantly, we learned several major lessons with regard to such an approach. Foremost, blocking of *pdhC* enhanced *Spn* virulence and growth rate; this is an outcome that should, of course, be avoided in vivo. It is therefore vital to broadly test and know the consequence of a

metabolic inhibiting agent on the bacteria's virulence. Moving forward, only specific pairings between drugs and metabolic inhibitors might be possible for treating bacterial diseases. Repurposing already FDA-approved drugs such as fomepizole could be used to enhance existing treatments for bacterial infection. Alternatively, a version of fomepizole with greater efficacy against the bacterial version of these metabolic enzymes might be possible to construct and would have better efficacy. This is an interesting option. In conclusion, our study, to our knowledge, is the first report demonstrating that drug inhibitor targeting fermentation pathways has potential as an adjuvant to antibiotics. We present our findings as proof-of-principle that blocking of NAD+ regeneration pathways during the fermentation has promise against drug-resistant gram-positive anaerobic pathogens and may be a novel therapeutic option.

## Materials and methods

### Bacterial culture

The bacterial strains used in the study are described in S1 Table; the prototype strain of *S. pneumoniae* used for the study was serotype 4 strain TIGR4 [24,50]. Construction of isogenic mutants was performed as described in a previous report [24]. Mutagenic PCR constructs were generated by amplifying the upstream and downstream DNA fragments flanking the gene(s) of interest, followed by the 5′ and 3′ integration of these fragments with the Janus cassette using a HiFi assembly master mix (NEB). Transformation of TIGR4 with the mutagenic construct (100 ng/mL) was induced using competence-stimulating peptide variant 2 (CSP-2). Kanamycin (300 mg/L) or streptomycin (150 mg/L) was used for the selection marker. The primers to construct the mutant and complementation panel used for this study are listed in S2 Table. For gene complementation, we transformed the amplicon using primer 1–6 pair with *Spn* WT chromosome, which amplified whole genes with 1K bp upstream and downstream. Each amplicon was transformed into a knockout strain and screened by kanamycin and streptomycin, as described previously, for complement strain construction [63,64]. All streptococcal cultures were prepared and cultured in Todd–Hewitt broth with 0.5% yeast extract (THY), or on tryptic soy blood agar plates (Remel), in a humidified atmosphere at 37°C with 5% $CO_2$. For growth curves, 4 independent cultures were prepared and optical density was measured during 6 hours of cultivation. Other than pneumococcus, *Streptococcus pyogenes* (GAS), *Streptococcus agalactiae* (GBS), and *E. faecium* were used for the study and prepared similarly with *Spn* [65]. Multidrug-resistant *Spn* serotype 35B was a gift from Dr. William Benjamin at UAB. Dr. Feroze Ganaie performed serotyping of this strain in Dr. Moon Nahm's laboratory at UAB using standard quelling methodology with capsule-specific antibody. We used bacteria in the early exponential phase of growth for all experiments. When we measured the impact of fomepizole to antibiotic susceptibility or energy metabolism, we added the drug to a final concentration of 50 mg/L, except where explicitly indicated.

### NAD+/NADH measurement

Measurement of NAD+/NADH was conducted using the NAD+/NADH-Glo assay (Promega #G9071) kit. We performed the assay following the manufacturer's instruction. For mutant panel sample preparation, mid-exponential culture was centrifuged down, and we subcultured to fresh THY media for 1 hour and analyzed its NAD+/NADH profile. To test the impact of fomepizole on the NAD+/NADH redox pool, we inoculated $5 \times 10^7$ CFU of exponential phase *Spn* in 50 ml THY and incubated with and without fomepizole for 2 hours, and the NAD +/NADH pool was examined as described above. A microplate reader (Cytation 5) was used for the luminescence.

## Intracellular ATP measurement

The measurement of ATP inside the bacteria was accomplished based on a previous report [66]. Briefly, mid-exponential phase *Spn* was treated with 0.1% trichloroacetic acid (TCA) for 30 minutes to extract its ATP. Extracted ATP solution was diluted in PBS 1:100 ratio to neutralize the acid and used for analysis. F2000 (Promega) ATP assay kit was used for measurement as manufacturer's instruction. A microplate reader (Cytation 5) was used to measure the luminescence. For the test, *Spn* WT and its isogenic mutants were prepared in the same manner as NAD+/NADH measurement.

## Antibiotic killing assay

The exponential phase of *Spn* TIGR4 WT and its isogenic mutants involved in the NAD + regeneration were inoculated in 6 ml THY media with different antibiotics. Approximately $1 \times 10^6$ CFU/ml were used to test, and the survival rate was measured at time point of 30 minutes, 60 minutes, 90 minutes, and 180 minutes. For each time point, CFU was determined using blood-agar plate (Remel). For the test, we added erythromycin (50 μg/ml), gentamicin (50 μg/ml), chloramphenicol (25 μg/ml), penicillin (1 μg/ml), vancomycin (10 μg/ml), and rifampicin (100 μg/ml) to the culture.

## Microdilution MIC assay

Mid-exponential phase bacteria were treated with either single antibiotics or antibiotics with fomepizole. About $5 \times 10^5$ CFUs/ml in mid-exponential phase bacteria were inoculated in a 96-well plate containing media with antibiotics and fomepizole. Three different groups of antibiotics targeting ribosomal activity, cell wall integrity and synthesis, and DNA transcription were used for the test. Erythromycin, gentamicin, kanamycin, penicillin, vancomycin, and rifampicin were used for the test. The chemicals and antibiotics were serially diluted with 2-fold dilution and incubated for 16 hours, and optical density was measured to determine the MIC. After measurement of optical density, resazurin was added with 0.003% concentration to detect bioactivity as described elsewhere [67].

## Immunoblot

*Spn* were grown to early exponential phase in THY. For the capsule immune dot blot assay, cultured *Spn* cells were collected and spotted on the nitrocellulose membrane with designated concentration within a 2-μL volume. As the primary antibody, rabbit anti-serotype 4 capsular polysaccharide antibody (no. 16747, Statens_Serum_Institut) was applied at a 1:20,000 ratio, and the mixture was incubated overnight at 4°C. As the secondary antibody, horseradish peroxidase (HRP)-conjugated goat anti-rabbit (AB_2307391; Jackson ImmunoResearch) was applied at a ratio of 1:10,000, and the mixture was incubated for 1 hour at room temperature. For pneumolysin measurement, rabbit anti-pneumolysin (Invitrogen) was used as primary antibody, and HRP-conjugated goat anti-rabbit was used as the secondary antibody-like capsule dot blot. Visualization was done using chemiluminescence via Clarity Western ECL enhanced chemiluminescence substrate (Bio-Rad).

## FITC-dextran exclusion assay for capsule measurement

To quantify the *Spn*'s capsule thickness, we measured the exclusion area of *Spn* within FITC-dextran (FD2000S; Sigma) [24,65]. *Spn* TIGR4 WT and its isogenic mutants were cultured in THY media until $OD_{600nm}$ 0.4 and centrifuged at 3,000$g$ for 10 minutes, and pellet was resuspended in 500 μl of PBS or 4% paraformaldehyde solution. For fomepizole sample, $1 \times 10^6$

CFU/ml concentration of *Spn* with and without fomepizole for 2 hours and prepared as described above. About 18 μl of resuspension was mixed with 2 μl FITC-dextran solution (10 mg/ml, final 1 mg/ml concentration). The mixed solution was put onto a microscope slide and visualized with Leica LMD6 microscope equipped with DFC3000G monochrome camera and Nikon A1R confocal microscope (UAB, HIRF). The obtained images were analyzed using ImageJ processing software.

### Animal infection

The competitive index was conducted as described previously [24]. For all animal studies, the inoculum was prepared at a 1:1 ratio mixture of *Spn* TIGR4 and its mutant culture for all animal studies. Screening for mutants was conducted using kanamycin as a selection marker. For the colonization model, $1 \times 10^6$ *Spn* in 10 μl was inoculated intranasally. Bacterial burden was examined by collecting nasal wash. For the pneumonia infection model, $1 \times 10^5$ *Spn* in 50 μl was inoculated intratracheally. At 48 hours post-infection, mice were killed, and harvested lung was homogenized for bacterial burden. To evaluate fomepizole's synergistic effects with antibiotics, we infected mice with $1 \times 10^6$ CFU of *Spn* serotype 35B 162–5678 intratracheally. After 18 hours of infection, we injected 100 μl of PBS (Control), antibiotic/compounds alone (Ery, concentration of 25 mg/kg and fomepizole, concentration of 50 mg/kg), or antibiotics with fomepizole intraperitoneally. At 48 hours post-infection point, lung, spleen, and heart were dissected from infected mice, and bacterial burden was analyzed.

### Ethics statement

All mouse experiments were reviewed and approved by the Institutional Animal Care and Use Committee at The University of Alabama at Birmingham, UAB (protocol no. IACUC-21851). Animal care and experimental protocols adhered to Public Law 89–544 (Animal Welfare Act) and its amendments, Public Health Services guidelines, and the *Guide for the Care and Use of Laboratory Animals* [68].

### Efflux capability measurement

The efflux capability of bacteria was measured as described previously with slight modification [69,70]. *Spn* were grown to mid-exponential phase, harvested, and concentrated in media to 2.0 optical density in PBS with 0.4% glucose and EtBr of 0.5 μg/ml concentration. The fluorescence level of EtBr was measured at 5 minutes intervals for 30 minutes using Cytation 5.

### Heat flow measurement

Microcalorimetric measurements were conducted using a calScreener according to the manufacturer's procedures and guidelines (Symcel, Sweden) and as previously described. Briefly, individual sterile plastic inserts were placed inside sterile titanium vials with forceps. Each vial was sealed with a titanium lid and tightened to identical torque (40 cNm) using a manufacturer-provided torque wrench. The tray containing 48 vials, consisting of 32 samples and 16 references (filled with sterile media), was inserted into the microcalorimeter (calScreener, Symcel, Sweden), and heat measurements were recorded with the software calView 1.033. The tray was preheated in position 1 for 10 minutes, moved to position 2 for 20 minutes, and then moved into the measuring chamber. The wells were stationary during all measurements, and the system was allowed to equilibrate for approximately 30 minutes before stable signals were recorded. Heat flow (in μW) measurements were recorded at a rate of 1 hertz.

## RT-qPCR

RT-qPCR was conducted to measure genes involved in NAD(H) metabolism using the previously described protocol [24]. The primers used for the test are listed in S2 Table. Bacterial culture was centrifuged and resuspended in RNAprotect bacterial reagent (Qiagen) and stored at −80°C [24]. Total RNA was prepared using the Rneasy plus kit (Qiagen, #74134) per the manufacturer's protocol without any modifications. The quantification of transcripts was accomplished with SYBR green qPCR master mix (Qiagen) using a C1000 thermal cycler (Bio-Rad).

## Statistics and data analysis

We performed a one-way analysis of variance (ANOVA) with Dunnett post hoc analysis for multiple data groups to compare with its control group. When we compared multiple data groups each other, we performed a one-way ANOVA with Turkey's post hoc analysis. If results are nonparametric, we performed a Kruskal–Wallis ANOVA test. A Mann–Whitney $t$ test was used to compare 2 independent groups or direct comparison with the control group. All tests in the study were conducted with at least triplicate biological replicates. Error bars on all main and supporting figures indicate standard deviation. This information is also available in each figure legend.

## Supporting information

**S1 Fig.** (**A**) Growth rate of the *Spn* TIGR4 and revertant strains in THY media. The growth rate was examined every hour for 6 hours ($n = 4$). (**B**) Intracellular ATP concentration of *Spn* TIGR4 and revertant strains ($n \geq 4$). The data underlying these figures can be found in S1 Data. Statistical analyses were done using the Mann–Whitney $t$ test and one-way ANOVA. (PDF)

**S2 Fig. Influence of NAD+ regeneration on *Spn* bioactivity.** Maximal heat flows of *Spn* wild type and its isogenic mutants involved in NAD+ regeneration during 24 hours of the cultivation. The measurement was made using the calorimeter. The data underlying this figure can be found in S1 Data. Statistical analyses were done using one-way ANOVA with the Dunnett post hoc test ($n \geq 3$). (PDF)

**S3 Fig. Competitive index of *Spn* TIGR4 and its mutant panel from nasopharyngeal colonization.** For nasopharyngeal colonization, mice were colonized intranasally $1 \times 10^6$ CFU of a 1:1 ratio mixture of *Spn* TIGR4 and mutant in 10 µl volume. The bacterial number was evaluated in the colonization model by a nasal wash at 2, 4, 7, 10, and 14 days after colonization. The number of mutants were enumerated using kanamycin as a selection marker. Statistical analyses were done using the Mann–Whitney $t$ test. The data underlying this figure can be found in S1 Data. Asterisks indicate statistical significance: *, $P \leq 0.05$, **, $P \leq 0.01$; ***, $P \leq 0.001$; ****, $P \leq 0.0001$. (PDF)

**S4 Fig. Erythromycin susceptibility of *Spn* TIGR4 and gene-complemented revertant strains.** The viability of *Spn* TIGR4 and revertant strains was measured after 90 and 180 minutes of exposure with and without erythromycin (50 µg/ml). The data underlying this figure can be found in S1 Data. ($n \geq 3$) Statistical analyses were done using the Mann–Whitney $t$ test. (TIFF)

**S5 Fig. Fomepizole alters pneumococcal capsule production and metabolic gene expression.** Fomepizole treatment decreased capsular polysaccharide production as determined by

(**A**) immuno dot blot using type 4 specific anti-capsule antibody. (**B**) Altered gene expression of NAD+ regeneration-associated genes. In comparing and analyzing the gene expression level, the qRT-PCR result was normalized using *rpoD* as a reference. For both tests, fomepizole was applied to bacteria for 2 hours. The data underlying S5B Fig can be found in S1 Data. Mann–Whitney *t* tests between fomepizole and control groups were done. Asterisks indicate statistical significance: *, $P \leq 0.05$, ***, $P \leq 0.001$.
(PDF)

**S6 Fig. Fomepizole treatment increased antibiotic susceptibility of *Spn*.** Antibiotic susceptibility was assessed using ribosome targeting antibiotics, erythromycin (5 μg/ml) and gentamicin (50 μg/ml), and cell wall synthesis inhibiting antibiotics, penicillin (1 μg/ml) and vancomycin (10 μg/ml) with and without fomepizole (FM). Cell viability was measured at the designated time points for 3 hours of incubation with antibiotics. The data underlying this figure can be found in S1 Data. Statistics were done with the Mann–Whitney *t* test between the antibiotics and a combination of fomepizole and antibiotic groups. Asterisks indicate statistical significance: ***, $P \leq 0.001$, ****, $P \leq 0.0001$.
(PDF)

**S7 Fig. Heat plots of microdilution checkerboard assay for combinations of fomepizole and antibiotics against *S. pneumoniae* serotype 2 D39, *S. pyogenes*, *S. agalactiae*, and *E. faecium*.** After 16–24 hours of incubation in THY media with designated amounts of fomepizole and antibiotics, viability of bacteria was assessed. At least 3 independent assays were performed ($n \geq 3$). The data underlying this figure can be found in S1 Data. For *Streptococcus*, erythromycin was used as a model antibiotic, and gentamicin was used for *E. faecium* due to its high resistance to other antibiotics.
(PDF)

**S8 Fig. DNA electrophoresis images of PCR amplicons from *Spn* TIGR4 and *Spn* 35B 162– 5678 strains.** Primers were used for PCR detection of antibiotic resistance genes *ermB*, *mef*(A/ E), and *aphA3* in the 162–5678 chromosome. White arrows indicated the correct band of the PCR products for each gene.
(PDF)

**S9 Fig. Growth rate, ATP generation, and antibiotic susceptibility of *adh*/*adhE* double knockout strain.** We generated the double knockout of the *adh*/*adhE* strain and evaluated its (**A**) growth rate, (**B**) intracellular ATP concentration, and (**C**) survival rate after erythromycin exposure. All assays were conducted as described in the Materials and methods. ($n > 3$) The data underlying this figure set can be found in S1 Data. Statistical analyses were done with the Mann–Whitney *t* test between TIGR4 WT. Asterisks indicate statistical significance: *, $P \leq 0.05$, **, $P \leq 0.01$; ***, $P \leq 0.001$; ****, $P \leq 0.0001$.
(PDF)

**S1 Table. Bacterial strains used in the study.**
(DOCX)

**S2 Table. Primers used in the study.**
(DOCX)

**S1 Data. Raw data for reproducing all main and supporting figures.** This file contains Excel tables organized by figure order, within which the raw result sets for each figure are displayed in separate sheets.
(XLSX)

**S1 Raw Images. Uncropped and unmodified blots and images corresponding to the main and supporting figures.** This file has the raw image of Figs 2A and 2B, S5 and S8. (PDF)

## Acknowledgments

Clinical isolate, *Spn* 35B 162–5678, was kindly given by Dr. William Benjamin at UAB. We thank Drs. Feroze Ganaie and Moon Nahm for the identification of its corresponding serotype. The calScreener for microcalorimetry analyses was provided by the company Symcel (Sweden). Mice images for Fig 6A were designed by Lemmling and obtained using image source: https://openclipart.org.

## Author Contributions

**Conceptualization:** Hansol Im, Carlos J. Orihuela.

**Funding acquisition:** Carlos J. Orihuela.

**Investigation:** Hansol Im, Madison L. Pearson, Kyle H. Cichos, Xiuhong Song, Katherine L. Kruckow.

**Methodology:** Hansol Im, Madison L. Pearson, Eriel Martinez, Kyle H. Cichos, Rachel M. Andrews, Elie S. Ghanem.

**Project administration:** Carlos J. Orihuela.

**Resources:** Carlos J. Orihuela.

**Supervision:** Carlos J. Orihuela.

**Validation:** Hansol Im, Madison L. Pearson, Eriel Martinez, Carlos J. Orihuela.

**Writing – original draft:** Hansol Im, Carlos J. Orihuela.

**Writing – review & editing:** Hansol Im, Madison L. Pearson, Eriel Martinez, Katherine L. Kruckow, Carlos J. Orihuela.

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
