## [Editor Report · Decision Letter 0]

6 Sep 2022

Dear Dr Orihuela, 

Thank you for submitting your manuscript entitled "Targeting NAD+ regeneration enhances antibiotic susceptibility of Streptococcus pneumoniae during invasive disease." for consideration as a Research Article by PLOS Biology.

Your manuscript has now been evaluated by the PLOS Biology editorial staff, as well as by an academic editor with relevant expertise, and I am writing to let you know that we would like to send your submission out for external peer review.

Once your full submission is complete, your paper will undergo a series of checks in preparation for peer review. After your manuscript has passed the checks it will be sent out for review. To provide the metadata for your submission, please Login to Editorial Manager (https://www.editorialmanager.com/pbiology) within two working days, i.e. by Sep 08 2022 11:59PM.

Kind regards,

Paula

Senior Editor

PLOS Biology

---

## [Decision Letter · Decision Letter 1]

18 Nov 2022

Dear Dr. Orihuela,

Thank you for your patience while your manuscript "Targeting NAD+ regeneration enhances antibiotic susceptibility of Streptococcus pneumoniae during invasive disease." was peer-reviewed at PLOS Biology. It has now been evaluated by the PLOS Biology editors, an Academic Editor with relevant expertise, and by several independent reviewers. 

In light of the reviews, which you will find at the end of this email, we would like to invite you to revise the work to thoroughly address the reviewers' reports.

As you will see below, the reviewers find your work interesting but they find issues that would need to be solved before publication. We think it is important that you address the technical points from reviewer #1, in particular the need for complementation assays to validate gene deletion phenotypes. We think it is essential that you include control experiments as requested, and experimental revision to address points 4-6 of reviewer #1 and to points 1 and 2 of reviewer #2. 

Given the extent of revision needed, we cannot make a decision about publication until we have seen the revised manuscript and your response to the reviewers' comments. Your revised manuscript is likely to be sent for further evaluation by all or a subset of the reviewers.

**IMPORTANT - SUBMITTING YOUR REVISION**

*Re-submission Checklist*

*Published Peer Review*

*PLOS Data Policy*

*Blot and Gel Data Policy*

Sincerely,

Paula

---

Senior Editor

PLOS Biology

REVIEWS:

Reviewer #1: Molecular epidemiology and pathogenesis of pneumococcal infections.

Reviewer #2: Bacterial pathogenesis and antibiotic resistance.

Reviewer #1: This is an interesting and potentially important paper in view of the current increase in macrolide resistance in Streptococcus pneumoniae. The finding that blocking the redox-balance enhances pneumococcal susceptibility to antibiotics that inhibit ribosome functions, suggest the use of redox modulators, such a inhibitors of alcohol-dehydrogenases as antibiotic potentiators. However, the paper would be stronger and more convincing by addition of some control experiments.

Comments:

1. The main message is that inhibition of alcohol dehydogenase with 4-methylpyrazole sensitizes macrolide resistant pneumococci to erythromycin in an animal treatment model. It would be good to add some sentences about the current situation of beta-lactam and macrolide resistance in Streptococcus pneumoniae as these are the two main classes of antibiotics used to treat pneumococcal infections.

2. A panel of mutants were generated in a set of pneumococcal genes affecting redox balance. Since they were constructed by inserting a kanamycin resistance casette, polar effects on down stream genes cannot be excluded. To be sure that the phenotype is caused by inactivation of the target gene, complementation needs to be done. Also, the kanamycin resistance casette may cause a negative effect in competition experiments with the wild-type (lacking the casette). 

3. Is the erythromycin dose given to the animals similar to the dose given for humans?

4. Macrolide resistance can be caused by mechanisms such as modification of the target and/or efflux. It is not evident that the alcohol-dehydrogenase inhibitor potentiates erythromycin for both types of resistancies.. It is easy to determine which resistance genes are present in the clinical 35B isolate used in this study.

5. The mechanisms explaining the increased sensitivity of the redox mutants to ribosome acting antibiotics has not been elucidated in this paper. Can something more on this topic be added to the paper?

6. Pneumococci have two genes encoding alcohol dehydrogenases. What is the phenotype when both these genes have been inactivated?

7. Change ng/ml to microgram/ml for all. In Table 1 there is some mixup since I do not think D39 is resistant to erythromycin.

Reviewer #2: [1] The authors provide a very brief explanation as to why capsule size is reduced in all of the mutants, but this could do with some more detail and expansion. It is taken as read that reduced capsule size means less virulent, but it would be interesting to test each of the mutants in both IV blood infection and upper respiratory tract colonisation models. Reduction in capsule size may not necessarily mean reduced survival and or colonisation rate. 

[2] The pneumolysin - lactate dehydrogenase connection is also interesting. Could the authors expand on this? What are the clinical implications for example? 

[3] Figure 5, lung CFU data is interesting. There are clearly two distinct CFU groups in control group of mice, one clustered around Log 4 another at Log 7. Do the authors have an explanation for this? I have not seen this type of spread in CFU before. Is it possible that that the lower CFU clustering represents the slower growing lung CFU from the initial intratracheal admin and the higher CFU cluster, those bacteria which came back in from blood infection? It would have been good to see blood CFU data over time to see when the bacteria translocated across to blood and how CFUs behaved there over time.

---

## [Decision Letter · Decision Letter 2]

27 Jan 2023

Dear Dr. Orihuela,

Thank you for your patience while we considered your revised manuscript "Targeting NAD+ regeneration enhances antibiotic susceptibility of Streptococcus pneumoniae during invasive disease." for publication as a Research Article at PLOS Biology. This revised version of your manuscript has been evaluated by the PLOS Biology editors, the Academic Editor and one of the original reviewers.

Based on the reviews and our Academic Editor's assessment of your revision, we are likely to accept this manuscript for publication, provided you satisfactorily address the remaining points raised by the reviewers regarding the corrections in Table 1. Please also make sure to address the following data and other policy-related requests.

1. DATA POLICY:

A) Supplementary files (e.g., excel). Please ensure that all data files are uploaded as 'Supporting Information' and are invariably referred to (in the manuscript, figure legends, and the Description field when uploading your files) using the following format verbatim: S1 Data, S2 Data, etc. Multiple panels of a single or even several figures can be included as multiple sheets in one excel file that is saved using exactly the following convention: S1_Data.xlsx (using an underscore).

B) Deposition in a publicly available repository. Please also provide the accession code or a reviewer link so that we may view your data before publication.

Regardless of the method selected, please ensure that you provide the individual numerical values that underlie the summary data displayed in the following figure panels as they are essential for readers to assess your analysis and to reproduce it: Figures 1BCD, 2AC, 3, 4, 5BCD, 6BCD, and supplementary figures SF1AB, SF2, SF3, SF4, SF5B, SF6, SF7, SF9ABC.

**Please also ensure that figure legends in your manuscript include information on where the underlying data can be found, and ensure your supplemental data file/s has a legend.**

We require the original, uncropped and minimally adjusted images supporting all blot and gel results reported in an article's figures or Supporting Information files. We will require these files before a manuscript can be accepted so please prepare and upload them now. We expect this for figure 2B.

Please carefully read our guidelines for how to prepare and upload this data: https://journals.plos.org/plosbiology/s/figures#loc-blot-and-gel-reporting-requirements

We expect to receive your revised manuscript within two weeks.

*Published Peer Review History*

*Press*

Sincerely,

Paula

---

Senior Editor,

pjaureguionieva@plos.org,

PLOS Biology

Reviewer remarks:

Reviewer #1: The revised version of the manuscript has addressed all previosus comments. I still think however that Table I needs revison.

---

## [Editor Report · Decision Letter 3]

2 Feb 2023

Dear Dr. Orihuela,

Thank you for the submission of your revised Research Article "Targeting NAD+ regeneration enhances antibiotic susceptibility of Streptococcus pneumoniae during invasive disease." for publication in PLOS Biology. On behalf of my colleagues and the Academic Editor, Tobias Bollenbach, I am pleased to say that we can in principle accept your manuscript for publication, provided you address any remaining formatting and reporting issues. These will be detailed in an email you should receive within 2-3 business days from our colleagues in the journal operations team; no action is required from you until then. Please note that we will not be able to formally accept your manuscript and schedule it for publication until you have completed any requested changes.

PRESS

Sincerely, 

Paula 

---

Senior Editor

PLOS Biology
